# Diagnostic Accuracy of Radiologic Methods for Ankle Syndesmosis Injury: A Systematic Review and Meta-Analysis

**DOI:** 10.3390/jcm8070968

**Published:** 2019-07-03

**Authors:** Dong-il Chun, Jae-ho Cho, Tae-Hong Min, Young Yi, Su Yeon Park, Kwang-Hyun Kim, Jae Heon Kim, Sung Hun Won

**Affiliations:** 1Department of Orthopaedic Surgery, Soonchunhyang University Hospital Seoul, 59, Daesagwan-ro, Yongsan-gu, Seoul 04401, Korea; 2Department of Orthopaedic Surgery, Chuncheon Sacred Heart Hospital, Hallym University, 77, Sakju-ro, Chuncheon-si, Gangwon-do 200-704, Korea; 3Department of Orthopaedic Surgery, Seoul Foot and Ankle Center, Inje University, 85, 2-ga, Jeo-dong, Jung-gu, Seoul 100-032, Korea; 4Department of Biostatistics, Soonchunhyang University Hospital Seoul, 59, Daesagwan-ro, Yongsan-gu, Seoul 04401, Korea; 5Department of Urology, Soonchunhyang University Hospital Seoul, 59, Daesagwan-ro, Yongsan-gu, Seoul 04401, Korea

**Keywords:** ankle, syndesmosis injury, ligament, diagnosis

## Abstract

Misdiagnosis and inadequate treatment of syndesmosis could result in significant long-term morbidity including pain, instability, and degenerative changes of the ankle joint. The objective of this systematic review and meta-analysis was to determine whether radiologic tests accurately and reliably diagnose ankle syndesmosis injury. Medline, Embase, and Cochrane were searched. The database search resulted in 258 full text articles that we assessed for eligibility, we used eight studies that met all the inclusion criteria. In subgroup meta-analysis, the sensitivity analysis showed significant differences only in the MRI (Magnetic Resonance Imaging), and specificity was not statistically significant. In diagnostic meta-analysis, the pooled sensitivity and specificity were 0.528 and 0.984 for X-rays, 0.669 and 0.87 for CT (Computed Tomography), and 0.929 and 0.865 for MRI, all respectively. For sensitivity, MRI showed significantly sensitivity as higher than the other methods, and we detected no significance for specificity. Syndesmosis injuries differed significantly in the accuracy of radiological methods according to the presence of accompanied ankle fractures. In patients with fractures, simple radiography has good specificity, and CT and MRI have high sensitivity and specificity irrespective of fracture; in particular, MRI has similar accuracy to gold standard arthroscopic findings.

## 1. Introduction

It is estimated that syndesmosis injuries account for 1%–18% of all ankle sprains, although the true incidence may be higher due to misdiagnosis in those with subtle diastasis [1,2,3,4,5,6]. Misdiagnosis and inadequate treatment of syndesmosis could result in significant long-term morbidity including pain, instability, and degenerative changes of the ankle joint [3,6,7,8]. Therefore, there are many clinical examinations and radiologic tests that can diagnose syndesmosis.

Clinical examinations that assess ankle syndesmosis include palpation of the tibiofibular ligaments [8,9,10], dorsiflexion with external rotation stress test [3,11], the squeeze test [6,12], Cotton test [10,13], and fibula translation test [11,12,13,14]. However, these clinical examinations are still questioned in terms of reliability and accuracy, and there is a limit to diagnosing ankle syndesmosis based only on these tests. In addition to clinical examination, a number of radiologic and surgical tests have been investigated for diagnosing ankle syndesmosis. In many studies, arthroscopic diagnosis is considered a gold standard [11,15,16]; however, it is expensive and invasive and therefore is usually performed not for diagnosis but for treatment of syndesmosis [17]. Of the radiographic tests, X-ray is widely used, especially to rule out fractures, but the reliability and validity of X-ray to diagnose ankle syndesmosis is still questionable [17,18]. Computed tomography (CT) has been shown to be more sensitive than X-ray for detecting ankle syndesmosis [19], but there is concern about high radiation doses. Ultrasonography (USG) is less expensive, faster, and it enables soft tissue patho-dynamic assessment; in addition, it does not expose the patient to radiation. However, it is difficult to acquire skill in USG; the technique has a stiffer learning curve, and reproducibility may be difficult. Magnetic Resonance Imaging (MRI) has been found to have the highest specificity and sensitivity, similar to that of arthroscopy [16,20,21,22], but the costs for these examinations are still relatively high. Although several studies have reported the reliability and accuracy of each test, there has been no comprehensive study of these tests.

Sman et al. [23] reported on a systematic review of clinical examinations for diagnosing ankle syndesmosis injury, and they reported that one clinical examination is more likely to be missed than many; thus, they recommend performing multiple clinical examinations and additional radiologic evaluations for accurate diagnoses. However, there is no systematic review or meta-analysis of radiological diagnostic tests for ankle syndesmosis, likely because there are many types of equipment, the measurement variables are not formalized, and it is difficult to compare them.

Hence, we conducted a systematic review and meta-analysis of each radiologic diagnostic test and method based on the syndesmosis area for accurately and reliably diagnosing ankle syndesmosis.

## 2. Experimental Section

We performed a systematic review and meta-analysis without language restrictions in accordance with the preferred reporting items for systematic review and meta-analyses statement.

### 2.1. Searching Strategy

We conducted a cross-search of all related literature in MEDLINE through March 2017 and used an optimally sensitive Cochrane Collaboration search strategy using MeSH headings for both anatomic and radiologic terms: “Inferior tibiofibular joint,” “joint, inferior tibiofibular,” “tibiofibular joint, inferior,” “articulation talocruralis,” “syndemos*,” “tibiofibular ankle syndesmosis,” “ankle syndesmosis, tibiofibular,” “syndesmosis, tibiofibular ankle,” “ankle syndesmosis,” “syndesmosis, ankle,” “tibiofibular syndesmosis,” “syndesmosis, tibiofibular,” “radiograph*,” “tomography, X-ray computed,” “computed X-ray tomography,” “X-ray computed tomography,” “tomography, spiral computed,” “compute tomography spiral,” and “spiral computed tomography.” We also searched EMBASE from 1978 to March 2017 and the Cochrane Library for studies that met the following criteria: (1) All adult patients who had results of radiologic evaluation for syndemosis regardless of the method and (2) studies that reported accurate measurements. The exclusion criteria were: (1) Studies on lateral ankle sprains, (2) animal or cadaver studies, and (3) review articles.

### 2.2. Data Collection and Analysis

The initial screening test of the electronic databases for study selection was based on information in the title and abstract. Two of the authors independently selected all articles by following the above criteria while assessing their quality, and all authors discussed the studies before final selection, including to resolve any disagreements. In cases of insufficient data, we reviewed the full text of the articles for future information that was reported that could be converted to the required values. We carefully cross-checked the references and data for each included study to ensure that no overlapping data were present and to maintain the integrity.

### 2.3. Statistical Analyses

We conducted the meta-analysis using “metafor” (Reference: Conducting Meta-Analyses in R with the metafor Package, https://www.jstatsoft.org/article/view/v036i03) and “mada” (Reference: Meta-Analysis of Diagnostic Accuracy with mada, https://cran.r-project.org/web/packages/mada/vignettes/mada.pdf) packages (R version 3.4.1). To calculate the effect size of sensitivity and specificity, we used the Freeman–Tukey double arcsine transformed proportion (PFT) to solve the bias problem with 0-cell entries. We used a random-effects model for the meta-analysis because the different studies had different cut-off values. We also tested for heterogeneity and considered I^2^ > 80% heterogeneous. Because the number of meta-analyses was insufficient, we performed a cumulative meta-analysis, which is one of the methods to visually confirm publication bias by examining effect size patterns by equipment and year order. We also conducted subgroup analyses to evaluate the role of methodological quality in the primary meta-analysis estimates. Subsequently, we compared each instrument to demonstrate their measurement differences. All results were reported with 95% confidence intervals (CI), and we considered the equipment effect of an indicator to be zero difference if the 95% CI contained a 0.

For meta-analysis of diagnostic accuracy, we used a hierarchical summary receiver operating characteristics (HSROC) model and a bivariate model to fit a summary ROC curve and to obtain summary estimates of sensitivity and specificity. We also performed bivariate meta-regression using a likelihood ratio test to compare the diagnostic accuracy between the three clinical instruments and conducted a post hoc analysis adjusting p with the Bonferroni correction method. We used a proportional hazards model to calculate the area under the SROC curve (AUC) and partial AUC (pAUC), and to enhance our conclusions, we used SROC curves to graphically present the diagnostic accuracy of the subgroups.

### 2.4. Assessing the Risk of Bias in the Included Studies

Two authors independently assessed the methodological quality of the studies and the data extraction, and discrepancies were resolved by consensus. We assessed risk of bias using the Quality in Prognosis Studies (QUIPS) tool.

### 2.5. Synthesis of Included Studies

We calculated sensitivity, specificity, diagnostic odds ratios, likelihood ratios, and positive and negative prediction values with 95% CIs. We performed subgroup meta-analyses by test and compared each diagnostic test.

## 3. Results

### 3.1. Inclusion of Studies

The initial search identified a total of 7607 articles from electronic databases (MEDLINE, 3401; Cochrane, 209; EMBASE, 3997). After we eliminated 2282 studies that contained overlapping data or that appeared in more than one database, and following the title and abstract screening, we selected 258 articles for possible inclusion in the review and retrieved the full-text articles. Of these, we eliminated 81 studies relating to lateral ankle sprains, 11 studies that reported results for cadavers, 42 review articles, and 114 studies that did not include accuracy results. We eventually used 8 studies [16,21,24,25,26,27,28,29] that met all the inclusion criteria and used an ultimate total of 6 [16,21,25,27,28,29], eliminating 2 studies [24,26] in meta-analysis because one used MRI as a reference and the other was the only study that used USG. A detailed flow chart of the selection process is shown in Figure 1. The included 8 studies contained detailed research duration and subject description information (Table 1); the research durations ranged from 1995 to 2017. 

### 3.2. Quality Assessment and Reporting Bias

Table 2 shows the quality assessment of the included studies using the QUIPS tool. All of the study authors reported the detailed reasons for their selected populations and included detailed descriptions of their sampling and measurement methods.

### 3.3. Outcome and Findings for Diagnostic Accuracy

Detailed outcomes and diagnostic accuracy findings of included studies are described in Table 3. Subgroup meta-analyses are described in Figure 2. Figure 2 also shows the cumulative meta-analysis findings. Using the estimates from the subgroup analysis, we conducted post hoc analyses of the mean differences between the three groups, adjusting the results with Bonferroni correction. Sensitivity analysis showed stochastically significant differences only for MRI (X-ray = CT < MRI, *p* for X-ray vs. CT = 1, *p* for X-ray vs. MRI < 0.001, and *p* for CT vs. MRI = 0.002); specificity was not statistically significant in the three groups (MRI = CT = X-ray; *p* for MRI vs. CT =1, *p* for MRI vs. X-ray =0.075, *p* for CT vs. MRI = 0.193). We show comparison results for each diagnostic test in Figure 3 and Table 4. We excluded studies that did not use intraoperative or arthroscopic findings as references in the meta-analysis, and furthermore, and meta-analysis involved the most valuable method in each study. Detailed outcomes and findings for each subgroup follow.

#### 3.3.1. X-ray

We included a total of three X-ray studies [16,21,26], and of these, Schoennagel et al. [26] reported on the accuracy of tibiofibular clear space and medial clear space for diagnostic accuracy compared with MRI, Takao et al. [16] reported on the accuracy of tibiofibular clear space, tibiofibular overlap, talar tilt, and medial clear space compared with arthroscopic findings, and Oae et al. [21] addressed tibiofibular clear space compared with arthroscopic findings.

We included two studies [16,21] in the meta-analysis that used random-effects models, and in the sensitivity analysis, the test results for heterogeneity were Q = 2.43, *p* = 0.129, and I^2^ = 58.92% and Q = 0.001, *p* = 0.0.98 and I^2^ = 0% for specificity. We calculated summary statistics for X-ray sensitivity and specificity and found 0.536 (0.319–0.747) and 0.984 (0.927–1), respectively (Figure 2).

#### 3.3.2. CT

We included two studies of CT [27,29]. Of these, Yeung et al. [27] reported on the diagnostic accuracy of tibiofibular distance divided into three points (anterior, middle, and maximum) compared with intraoperative findings, and Ahn et al. [29] reported the narrowest tibiofibular distances compared with arthroscopic findings. We used a random-effects model for the CT studies, and the I^2^ values for sensitivity and specificity were 73.88% (Q = 3.83 and *p* = 0.05) and 62.1% (Q = 2.64 and *p* = 0.1), respectively. The mean sensitivity and specificity for CT in the two studies were 0.67 (0.468–0.844) and 0.984 (0.927–1), also respectively (Figure 2)

#### 3.3.3. MRI

We included four studies on MRI [16,21,25,28]. Takao et al. [16] reported on syndesmosis tear following three cases following discontinuity, decrease of tension, and abnormal ligament course for diagnostic accuracy compared with arthroscopic findings. Oae et al. [21] reported on four syndesmosis cases that included ligament discontinuity, wavy, curved ligament contours, and no visualization of ligament, and Clanton et al. [28] reported on three cases with ligament tear, sprain, and scarring for accuracy, also compared with arthroscopic findings. Kim et al. [25] reported on both routine and contrast-enhanced MRI divided into a five-point scoring system for syndesmosis compared with arthroscopic findings and defined scores of more than 4 as syndesmosis. We used these four studies to estimate summary statistics based on a random-effect model and found sensitivity of 0.963 (0.908–0.997) and specificity of 0.875 (0.729–0.973) with no significant heterogeneity (sensitivity: Q = 2.18, *p* = 0.54, and I^2^ = 0%; specificity: Q = 3.49, *p* = 0.32, and I^2^ = 23.2%; Figure 2)

#### 3.3.4. USG

One study with ultrasound met our inclusion criteria [24]. This study reported sensitivity and specificity of USG for diagnosis of syndesmosis of 0.886 (0.755–0.974) and 0.967 (0.886–1), respectively. However, there was only one study with ultrasound, and we excluded this study from the meta-analysis.

#### 3.3.5. Diagnostic Accuracy for X-ray, CT, and MRI

In the diagnostic meta-analysis, we analyzed six studies (Table 4) [16,21,25,27,28,29]. To solve the cut-off value problems, we used the HSROC model to evaluate and compare clinical instruments. The pooled sensitivity and specificity (1- false positive rate) were 0.528 and 0.984 for X-rays, 0.669 and 0.87 for CT, and 0.929 and 0.865 for MRI, all respectively. For sensitivity, MRI showed significantly higher accuracy than the other methods (X-ray = CT < MRI), and we detected no significance for specificity. We calculated AUC and pAUC for each method, and the summary values were for 0.922 and 0.817 for X-ray, 0.877 and N/A for CT, and 0.834 and 0.678 for MRI, also all respectively (Table 4 and Figure 3). Although one limitation of our study is the limited number of studies we included in our analysis, we still found meaningful results even after we resolved the cut-off value problem.

## 4. Discussion

This current systematic review with meta-analysis regarding the issue of diagnostic accuracy of radiologic tests for syndesmosis injury has one salient point: To our best knowledge, our study was the first systematic review with meta-analysis of radiologic methods for diagnosis of ankle syndesmosis. There are many radiologic tests for this injury, and each test has many methods, and for this reason, it can be difficult to generalize the radiologic test findings and interpretations; we were able to overcome some of the heterogeneity by including only studies that reported accuracy measurements. In our study, the sensitivity of the diagnostic accuracy for ankle syndesmosis was as follows: MRI, 0.929; CT, 0.669; X-ray, 0.528. Additionally, specificity values were as follows: X-ray, 0.984; CT, 0.87; MRI, 0.865.

We included three studies [16,21,26] that used X-ray to diagnose syndesmosis injury in our systematic review, although of these, we excluded one study [26] from the meta-analysis because its authors used MRI for reference. The authors of these studies investigated two different anatomical areas, that of syndesmosis ligaments and non-syndesmotic areas. The authors investigated the tibiofublar clear space and tibiofubular overlap in the syndesmotic areas and the medial clear space and talar tilt in the non-syndesmotic areas. There was one important difference between the studies that we did and did not include in the meta-analysis: The specificity values were very different. In the studies we included, the specificity was as high as nearly 100%, whereas in the studies we did not include, it was as moderate as 59% to 75%. We suspect two reasons for this circumstance. First, the study populations were different; those we included in the meta-analysis involved patients with accompanying ankle fractures patients, and the studies we did not include investigated patients with isolated syndesmosis injuries without fractures. The second reason is that arthroscopic findings were used as a reference in the studies we included in the meta-analysis, but MRI was used in the studies we did not include. For these reasons, our meta-analysis showed high specificity for using X-ray to diagnose syndesmosis; specificity is calculated according to the true negative (TN)/false positive (FP) + TN formula. In the studies we included in the meta-analysis, FP was zero; in other words, if syndesmosis injury was seen on the X-ray, it can be interpreted that there was no exception of syndesmosis injury in patients with accompanying ankle fracture. However, in the study that did not involve patients with accompanying fracture, we observed high FP. Therefore, we conclude that the X-ray has excellent specificity for diagnosis of syndesmosis injury in patients with accompanying fracture, but sensitivity is moderate, and has moderate sensitivity and specificity both for isolated syndesmosis injury without fracture.

We included two studies [27,29] that used CT to diagnose syndesmosis injury in our systematic review and meta-analysis. Yeung et al. [27] reported that anterior tibiofibular distance and maximal tibiofibular distance in the axial CT scan were the most powerful predictors of positive syndesmosis injury in patients with accompanying fracture, and Ahn [29] reported that the narrowest tibiofibular distance in the axial CT scan had a positive correlation with acute or chronic syndesmosis injury in patients without ankle fracture. In our meta-analysis, CT had good sensitivity and excellent specificity for diagnosis of syndesmosis injury, but the radiation from CT is still a concern among both physicians and patients. Therefore, we conclude that CT has excellent specificity and good sensitivity for diagnosing ankle syndesmosis injury regardless of accompanying fracture but there are concerns about radiation exposure.

We included in our systematic review four studies [16,21,25,28] that used MRI to diagnose syndesmosis injury, and of these, Kim et al. [25] compared routine versus contrast-enhanced MRI, and we included only one study that incorporated contrast-enhanced MRI in our meta-analysis. There were many ways to diagnose syndesmosis injury on MRI, but it was a common finding that authors of each study focused on the shapes and signals of the ligaments. In our meta-analysis, MRI had excellent sensitivity and specificity for diagnosis of syndesmosis injury regardless of accompanying fracture. Therefore, with proper and standard interpretations, MRI can be a standard method for comparing arthroscopic with intraoperative findings for syndesmosis injury.

We also included one study [24] that used USG to diagnose syndesmosis injury in our systematic review: Christo et al. [24] reported excellent sensitivity and specificity for accurate diagnosis of syndesmosis injury using USG at the interosseous membrane of the syndemosis area. However, we think that one limitation of our study is that we could not conduct a proper meta-analysis because of the lack of USG-based studies based on our selection criteria.

Although this study has the attribute of being the first diagnostic systematic review and meta-analysis of syndesmosis injury, there were still several limitations. First, we only included a few studies, primarily because our inclusion criteria required only studies that reported accuracy measurements, and thus we excluded many clinical studies on the diagnosis of syndemosis injury. However, because there were too many radiologic methods for diagnosis of syndesmosis injury, it was an unavoidable methodological choice to perform a standardized systematic review and meta-analysis. Second, we did not include prospective studies on the diagnosis of syndesmosis injury in our study because there were too few related studies; we think that this could be a limitation of the surgery department. Third, we did meta-analysis involving syndesmosis injury with ankle fractures, not only without fracture type. Of course, syndesmosis injuries with and without fracture are quite different entities. Syndesmosis injury without ankle fracture was more difficult to diagnosis compared with syndesmosis injury accompanied with ankle fracture. Syndesmosis injury accompanied ankle fracture type also was difficult to diagnosis. Therefore, we thought our results of both types were meaningful. Fourth, we could not involve the weight bearing CT scan. Of course, weight bearing CT could help to diagnose syndesmosis injury, however, few studies reported the effectiveness of the weight bearing CT. Therefore, we excluded the weight bearing CT in our study. We think more studies are needed when it comes to the role of the weight bearing CT for diagnosis of syndesmosis injury. Fifth, although we used the random-effects model for the meta-analysis to overcome the heterogeneity of each of the studies, we could not overcome it completely. This is thought to be due to the use of various tools in the diagnosis of an ankle syndesmosis injury, and a more delicate future study will be needed.

## 5. Conclusions

In conclusion, this is the first systematic review with meta-analysis that investigated the accuracy of radiologic tests for diagnosis of ankle syndesmosis injury. Although based on our study we still cannot clearly say what method is best for diagnosis, appropriate diagnostic methods based on the patient’s condition are not likely to miss the diagnosis of syndesmosis injury. Specifically, in patients with ankle fracture, simple radiography has good specificity, and CT and MRI have high sensitivity and specificity irrespective of accompanying fracture. In particular, MRI has similar accuracy to gold standard arthroscopic findings. Therefore, physicians should be careful not to miss the syndesmosis injury by closely examining simple radiographs in patients with fractures.

## Figures and Tables

**Figure 1 jcm-08-00968-f001:**
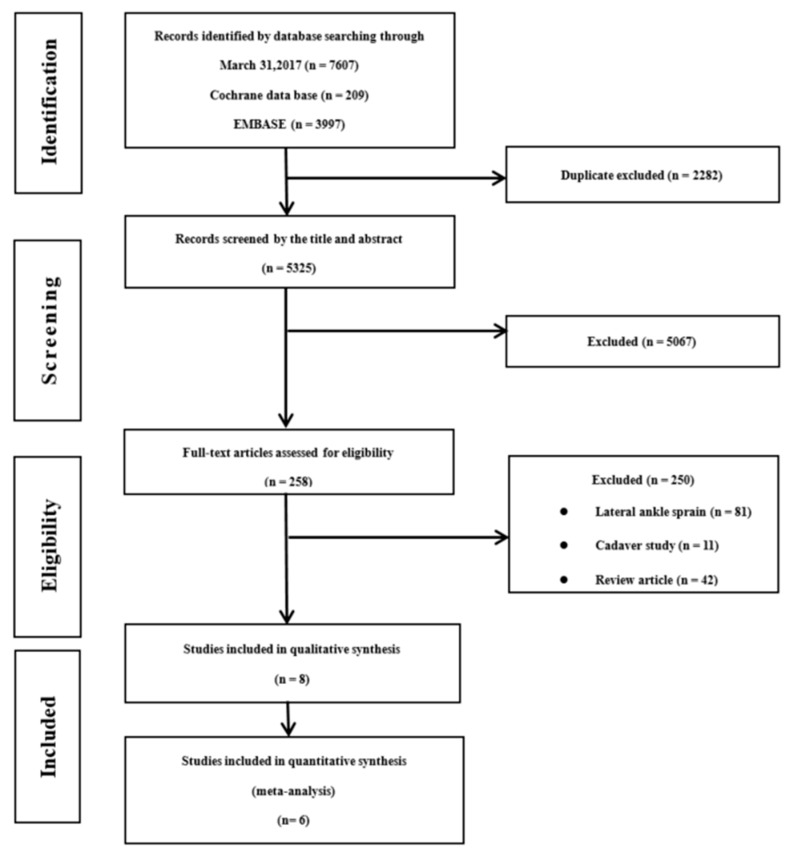
Flow chart of the included studies.

**Figure 2 jcm-08-00968-f002:**
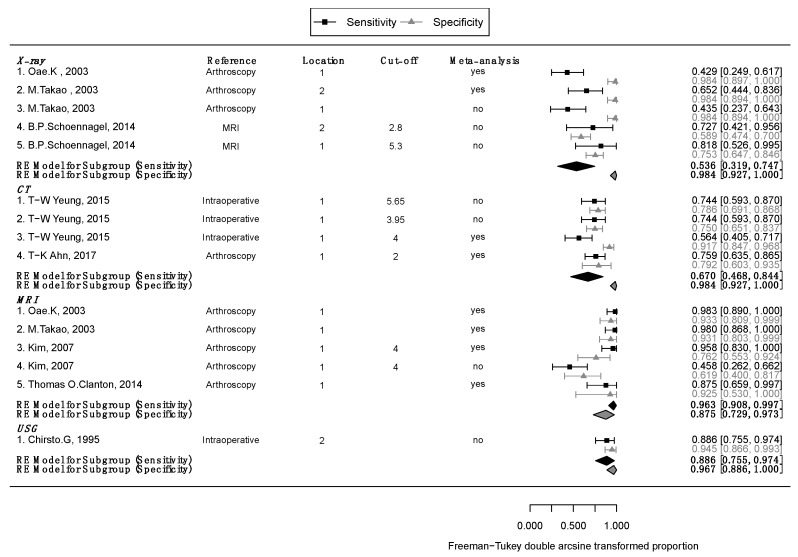
Subgroup meta-analysis and cumulative meta-analysis of each diagnostic methods.

**Figure 3 jcm-08-00968-f003:**
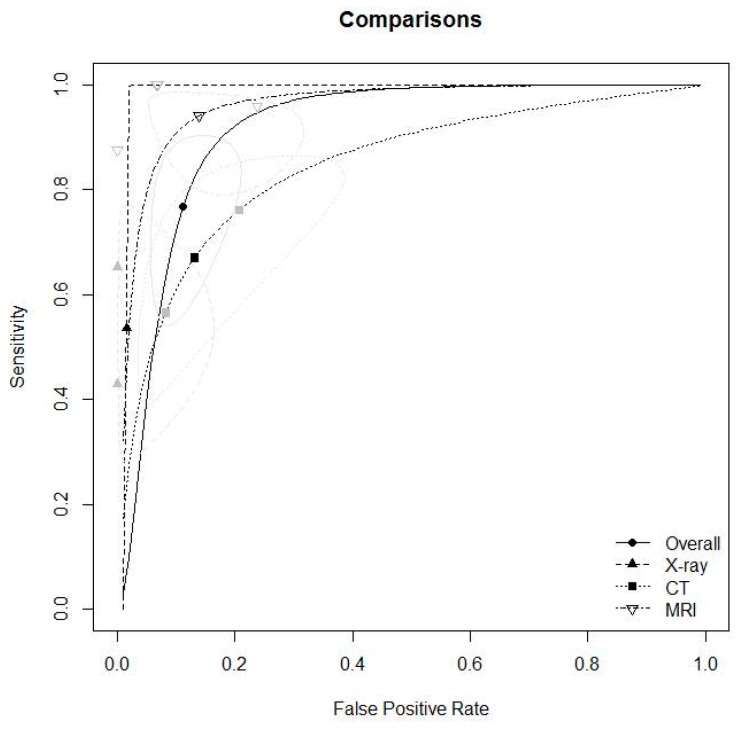
Comparison results for each diagnostic method.

**Table 1 jcm-08-00968-t001:** Characteristics and results of the included studies.

Author (year)	Location	No. of Samples	Median Age (Range)	Test	Fracture	Reference	
B.P.Schoennagel (2014)	Germany	84	32.8 ± 9.9	X-ray	no	MRI	
M.Takao (2003)	Japan	52	35 (14–67)	X-ray	yes	Arthroscopy	
Oae.K (2003)	Japan	58	37.4 (12–79)	X-ray	yes	Arthroscopy	
T-K Ahn (2017)	Korea	78	34.2 (15–64)	CT	no	Arthroscopy	
T-W Yeung (2015)	Hongkong	123	43.2 (11–82)	CT	yes	Intraoperative findings	
S.J.Kim (2007)	Korea	45	32.1 (18–58)	MRI	no	Arthroscopy	
M.Takao (2003)	Japan	52	35 (14–67)	MRI	yes	Arthroscopy	
Oae.K (2003)	Japan	58	37.4 (12–79)	MRI	yes	Arthroscopy	
T.Clanton (2014)	USA	21	35 (16–60)	MRI	no	Arthroscopy	
Chirsto.G (1995)	Greece	90	36.5 (18–80)	USG	yes	Intraoperative findings	

**Table 2 jcm-08-00968-t002:** Risk of bias for included studies.

Author (Year)	Location	Detailed Reasons for Selected Population	Detailed Description of Sampling and Measurement Method	Study Participation	Outcome Measurement
B.P. Schoennagel, 2014	Germany	Adult patients with acute ankle trauma that presented in our emergency unit	Yes	low 5	low 3
M.Takao,2003	Japan	Patients with acute injuries of the ankle which were treated surgically	Yes	moderate 4	low 3
Oaek,2003	Japan	Patients with ankle sprains and distal fibular fractures underwent surgery and ankle arthroscopy	Yes	moderate 4	low3
T.K. Ahn,2017	Korea	Who underwent an arthroscopic examination for the syndesmosis injury	Yes	low 6	low 3
T.W.Yeung, 2015	Hongkong	Patients who received CT scan for preoperative assessment with subsequent operation performed were in	Yes	low 5	low 3
S.J. Kim,2007	Korea	Who had a history of repeated ankle sprain and who had undergone MRI and arthroscope surgery	Yes	moderate 4	low 3
M.Takao,2003	Japan	Patients with acute injuries of the ankle which were treated surgically	Yes	moderate 4	low 3
Oae K,2003	Japan	Patients with ankle sprains and distal fibular fractures underwent surgery and ankle arthroscopy	Yes	moderate 4	low 3
T. Clanton, 2014	USA	Who underwent arthroscopically assisted surgery for ankle pathology and suspected syndesmosis injury	Yes	low 6	moderate 2
Chirsto.G,1995	Greece	Who had Weber type B and C ankle fracture	Yes	low 5	low 3

**Table jcm-08-00968-t003a:** 

Author	Test	Location	Method	Reference	Unit	Cut-off	TP	FN	FP	TN
B.P. Schoennaqel, 2014	X-ray	1	T-F clear	MRI	mm	5.3 mm	9	2	18	55
B.P. Schoennaqel, 2014	X-ray	2	Med. clear	MRI	mm	2.8 mm	8	3	30	43
M.Takao, 2003^†^	X-ray	1	T-F clear, T-F overlap	Arthroscopy	mm		10	13	0	29
M.Takao, 2003^†^	X-ray	2	talar tilt, med. clear space	Arthroscopy	degree, mm		15	8	0	29
Oae.K, 2003^†^	X-ray	1	T-F clear	Arthroscopy	mm		12	16	0	30
T-K Ahn, 2017	CT	1	Narrowest T-F distance at joint	Arthroscopy	mm	2 mm	41	13	5	19
T-W Yeung, 2015	CT	1	Ant. T-F distance	Intraoperative findings	mm	4 mm	22	17	7	77
T-W Yeung, 2015	CT	1	Mid T-F distance	Intraoperative findings	mm	3.95 mm	29	10	21	63
T-W Yeung, 2015	CT	1	Max T-F distance	Intraoperative findings	mm	5.65 mm	29	10	18	66
S.J.Kim, 2007	MRI	1	1: AITFL was definitely not injured, 2: AITFL was probably not injured, 3: AITFL was possibly injured, 4: AITFL was probably injured, 5, that the AITFL was definitely injured. Thickening, wavy contour, redundancy, discontinuity, or absence of the ligament	Arthroscopy	Signal intensity, score	4	11	13	8	13
S.J.Kim, 2007	MRI	1	1: AITFL was definitely not injured, 2: AITFL was probably not injured, 3: AITFL was possibly injured, 4: AITFL was probably injured, 5, that the AITFL was definitely injured.Thickening, wavy contour, redundancy, discontinuity, or absence of the ligament	Arthroscopy	Signal intensity, score	4	23	1	5	16
M.Takao, 2003 ^†^	MRI	1	Discontinuity, decrease of tension, abnormal course of the ligament	Arthroscopy	Signal intensity		23	0	2	27
Oae.K, 2003 ^†^	MRI	1	Ligament discontinuity, wavy, curved ligament contour, no visualization of ligament	Arthroscopy	Signal intensity		28	0	2	28
Thomas O.Clanton, 2014 ^†^	MRI	1	Ligament tear, sprain, scarring	Arthroscopy	Signal intensity		14	2	0	5
Chirsto.G, 1995	USG	2	Interosseous membrane	Intraoperative findings	hyperechogenic		31	4	3	52

**Table jcm-08-00968-t003b:** 

DOR			posLR			negLR			PPV			NPV			meta
Estimate	95% CI		Estimate	95% CI		Estimate	95% CI		Estimate	95% CI		Estimate	95% CI		
	lower	upper		lower	upper		lower	upper		lower	upper		lower	upper	
11.400	2.571	50.544	3.167	1.940	5.168	0.278	0.091	0.844	0.333	0.250	0.417	0.965	0.956	0.974	No
3.464	0.918	13.071	1.719	1.092	2.705	0.496	0.201	1.223	0.211	0.158	0.263	0.935	0.917	0.952	No
45.889	2.502	841.714	26.250	1.619	425.584	0.572	0.401	0.817	0.955	0.929	0.980	0.686	0.622	0.750	No
107.588	5.816	1990.220	38.750	2.441	615.026	0.360	0.209	0.619	0.969	0.954	0.984	0.776	0.721	0.832	Yes
46.212	2.570	831.063	26.724	1.656	431.189	0.578	0.420	0.796	0.962	0.941	0.982	0.649	0.584	0.714	Yes
11.985	3.734	38.462	3.644	1.647	8.064	0.304	0.181	0.510	0.891	0.863	0.919	0.594	0.510	0.677	Yes
14.235	5.239	38.681	6.769	3.162	14.489	0.476	0.331	0.683	0.759	0.692	0.825	0.819	0.789	0.849	Yes
8.700	3.637	20.809	2.974	1.967	4.498	0.342	0.198	0.592	0.580	0.512	0.648	0.863	0.836	0.890	No
10.633	4.376	25.837	3.470	2.215	5.437	0.326	0.189	0.563	0.617	0.549	0.685	0.868	0.843	0.894	No
1.353	0.422	4.333	1.191	0.605	2.342	0.880	0.539	1.438	0.579	0.469	0.689	0.500	0.404	0.596	No
47.000	6.946	318.027	3.760	1.811	7.806	0.080	0.017	0.385	0.821	0.767	0.876	0.941	0.915	0.967	Yes
517.000	23.622	11315.436	11.750	3.581	38.556	0.023	0.001	0.354	0.904	0.870	0.937	0.982	0.976	0.989	Yes
649.800	29.851	14144.756	12.186	3.709	40.038	0.019	0.001	0.293	0.919	0.893	0.945	0.983	0.977	0.989	Yes
63.800	2.624	1551.248	10.235	0.715	146.493	0.160	0.050	0.517	0.967	0.950	0.983	0.688	0.539	0.836	Yes
134.333	28.183	640.302	16.238	5.369	49.110	0.121	0.048	0.305	0.912	0.885	0.939	0.929	0.911	0.946	No

† continuity correction applied to zero cells.TP, true positive; FN, false negative; FP, false positive; TN, true negative; DOR, diagnostic odds ratio; posLR, positive likelihood ratio; negLR, negative likelihood ratio; PPV, positive predictive value; NPV, negative predictive value; CI, confidence interval; T-F, tibiofibular; Med., medial; ant., anterior; mid, middle; max, maximum; AITFL, anterior inferior tibiofibular ligament.

**Table 4 jcm-08-00968-t004:** Comparison results for each diagnostic test; CI, confidential interval; AUC, area under the curve; pAUC, partial area under the curve.

Comparison Results	Total (n = 8)			X-ray (n = 2)			CT (n = 2)			MRI (n = 4)			p-value^₭^	post-hoc^₤^
	Estimate	95% CI		Estimate	95% CI		Estimate	95% CI		Estimate	95% CI			
		Lower	Upper		Lower	Upper		Lower	Upper		Lower	Upper		
Heterogeneity Test, Higgins’ I2(*p* value) †	16.8% (*p* = 0.274)			0% (*p* = 0.897)			0% (*p* = 0.906)			0% (*p* = 0.592)				
Sensitivity ^‡^	0.745	0.592	0.854	0.528	0.380	0.672	0.669	0.506	0.800	0.929	0.837	0.971	0.004	X-ray = CT<MRI
Specificity ^‡^	0.882	0.815	0.927	0.984	0.893	0.998	0.870	0.739	0.941	0.865	0.753	0.931		MRI=CT =X-ray
AUC ^₮^	0.922	0.907	0.938	0.877	0.798	0.974	0.834	0.716	0.999	0.995	0.957	1.000		
pAUC ^₮^	0.817	0.783	0.854	N/A			0.678	0.463	0.997	0.988	0.898	1.000		

^†^*p* value by Cochran’s Q Test; ^‡^ estimated by the bivariate model of HSROC; ^₭^ Statistical test for evidence of a difference between groups by using F-test; ^₤^ the bivariate-meta regression was performed and *p* value adjusted by Bonfferoni correction; ^₮^ estimated by proportional hazard model approach.

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
