# Peer review of "Diagnostic Accuracy of Radiologic Methods for Ankle Syndesmosis Injury: A Systematic Review and Meta-Analysis"

_jcm, 2019, doi:10.3390/jcm8070968_

Round 1
Reviewer 1 Report
This manuscript is about meta analysis of radiology diagnosis of ankle syndesmosis. The authors have searched extensively the related articles and finally included 6 of them for quantitative analysis of diagnostic sensitivity and specificity. They concluded that MRI has a higher sensitivity than either CT or X-ray for diagnosing ankle syndesmosis. There was no significant difference regarding their respective specificities. Radiography is specific if syndesmosis occurs together with fractures.
General comments:
1. Inclusion of a small number of papers for analysis is the weakest point. However, the authors are manage to use statistical tools to compute the data.
2. I would recommend the authors to describe MRI “sensitivity as higher than”….. and not to describe MRI showed “higher accuracy than other methods”.
3. There are typo errors in result section, “Sensitivity analysis showed stochastically……. for MRI (X-ray = CT < MRI,,,,,,,,, and p for CT vs. X-ray = 0.002)”. Do the authors mean CT vs. MRI = 0.002? Similarly, “specificity was not statistically…….. in the 3 groups (MRI = CT = X-ray,,,,,,, p for CT vs. X-ray=1.000). Please check.
4. My recommendation is that these p-values are better incorporated in either Table 4 or a another new table.
5. I suggest that the authors should discuss or comment on the heterogeneity of the effects of their selected papers.
Author Response
1. Inclusion of a small number of papers for analysis is the weakest point. However, the authors are manage to use statistical tools to compute the data.
Response: We appreciate your comment.
2. I would recommend the authors to describe MRI “sensitivity as higher than”….. and not to describe MRI showed “higher accuracy than other methods”.
Response: We appreciate your comment. We change “higer accuracy than other methods” to “sensitivity as higher than” on line 30 in Abstract,
3. There are typo errors in result section, “Sensitivity analysis showed stochastically……. for MRI (X-ray = CT < MRI,,,,,,,,, and p for CT vs. X-ray = 0.002)”. Do the authors mean CT vs. MRI = 0.002? Similarly, “specificity was not statistically…….. in the 3 groups (MRI = CT = X-ray,,,,,,, p for CT vs. X-ray=1.000). Please check.
Response: We appreciate your comment. We change type errors as you recommend. Line 153~156; Sensitivity analysis showed stochastically significant differences only for MRI (X-ray = CT < MRI, p for X-ray vs. CT = 1.000, p for X-ray vs. MRI < 0.001, and p for CT vs. MRI = 0.002); specificity was not statistically significant in the 3 groups (MRI = CT = X-ray; p for MRI vs. CT =1.000, p for MRI vs. X-ray =0.075, p for CT vs. MRI = 0.193).
4. My recommendation is that these p-values are better incorporated in either Table 4 or a another new table.
Response: We appreciate your comment. We changed Table 4 for better corporation as following;
5. I suggest that the authors should discuss or comment on the heterogeneity of the effects of their selected papers.
Response: We appreciate your comment. We totally agree with your opinion. However, due to variety of methods are used to diagnose ankle syndesmosis injury, studies included in our meta-analysis had a different method for proving of accuracy of diagnostic tools for ankle syndesmosis injury. Therefore, we thought that heterogeneity of meta-analysis inevitable. So, we used random-effects model for the meta-analysis because the different studies had different cut-off values. We add this limitation on discussion section; Fifth, although we used random-effects model for the meta-analysis to overcome the heterogeneity of each of the studies, we could not overcome completely. This is thought to be due to the use of various tools in the diagnosis of an ankle syndesmosis injury, and a more delicate future study will be needed.
Reviewer 2 Report
This is a very well-done written and an interesting review.
- Add few more references to support the data presented in your review
- Not all abbreviations are explained. Make sure to write the whole phrase the first time used then use the abbreviation.
Author Response
This is a very well-done written and an interesting review.
Response: We appreciate your comment.
- Add few more references to support the data presented in your review
Response: We appreciate your comment. We totally agree with your opinion. However, we only included a few studies, because our inclusion criteria required only studies that reported accuracy measurements (such sensitivity, specificity, positive predict value, negative predict value etc.), and thus we excluded many clinical studies on the diagnosis of syndemosis injury. However, because there were too many radiologic methods for diagnosis of syndesmosis injury, it was an unavoidable methodological choice to perform a standardized systematic review and meta-analysis.
- Not all abbreviations are explained. Make sure to write the whole phrase the first time used then use the abbreviation.
Response: We appreciate your comment. We checked again.
Round 2
Reviewer 1 Report
Interesting article. The authors have edited a new version of manuscript.